# Chronic presence of blood circulating anti-NMDAR1 autoantibodies impairs cognitive function in mice

William Yue[1], Sorana Caldwell[1,2,3], Victoria Risbrough[1,2,4], Susan Powell[1,2,3], Xianjin Zhou[1,2,3]*

1 Department of Psychiatry, University of California San Diego, La Jolla, California, United States of America, 2 VA Research Service, VA San Diego Healthcare System, San Diego, California, United States of America, 3 VA Mental Illness Research and Clinical Core, VA San Diego Healthcare System, San Diego, California, United States of America, 4 VA Center of Excellence for Stress and Mental Health, VA San Diego Healthcare System, San Diego, California, United States of America

* xzhou@ucsd.edu

**Data Availability Statement:** All relevant data are within the manuscript and its Supporting information files.

## Abstract

High titers of anti-NMDAR1 autoantibodies in brain cause anti-NMDAR1 encephalitis that displays psychiatric symptoms of schizophrenia and/or other psychiatric disorders in addition to neurological symptoms. Low titers of anti-NMDAR1 autoantibodies are reported in the blood of a subset of the general human population and psychiatric patients. Since ~0.1–0.2% of blood circulating antibodies cross the blood-brain barriers and antibodies can persist for months and years in human blood, it is important to investigate whether chronic presence of these blood circulating anti-NMDAR1 autoantibodies may impair human cognitive functions and contribute to the development of psychiatric symptoms. Here, we generated mice carrying low titers of anti-NMDAR1 autoantibodies in blood against a single antigenic epitope of mouse NMDAR1. Mice carrying the anti-NMDAR1 autoantibodies are healthy and display no differences in locomotion, sensorimotor gating, and contextual memory compared to controls. Chronic presence of the blood circulating anti-NMDAR1 autoantibodies, however, is sufficient to impair T-maze spontaneous alternation in the integrity of blood-brain barriers across all 3 independent mouse cohorts, indicating a robust cognitive deficit in spatial working memory and/or novelty detection. Our studies implicate that chronic presence of low titers of blood circulating anti-NMDAR1 autoantibodies may impair cognitive functions in both the general healthy human population and psychiatric patients.

## Introduction

N-methyl-D-aspartate receptor (NMDAR) in brain is essential for learning and memory as well as other cognitive functions. Pharmacological studies demonstrated that NMDAR antagonists cause schizophrenia-like symptoms in human [1]. Recent human genetic studies further validated the central role of NMDAR functions in the development of schizophrenia [2, 3]. In addition to genetic mutations, NMDAR functions can also be impaired by physiological/

**Funding:** This work was supported by the grants R21MH123705 (XZ) and R21MH116186 (XZ) from the National Institute of Mental Health.

**Competing interests:** The authors declare no conflict of interest.

environmental risk factors. High titers of anti-NMDAR1 autoantibodies in human brain cause anti-NMDAR1 encephalitis that exhibits psychosis, memory loss, and other prominent psychiatric symptoms in addition to neurological symptoms [4, 5]. In fact, anti-NMDAR1 encephalitis is often misdiagnosed as schizophrenia, further supporting impaired NMDAR functions in the pathogenesis of schizophrenia and other psychiatric disorders.

Low titers of anti-NMDAR1 autoantibodies are commonly detected in human blood. Several studies reported that ~5–10% of the human population, regardless of healthy persons or psychiatric patients, carries low titers of anti-NMDAR1 autoantibodies in their blood [6–8]. These anti-NMDAR1 autoantibodies inhibit NMDAR functions in both *in vitro* human neurons [9] and *in vivo* mouse models [7, 10]. Although peripheral circulating antibodies are largely blocked from entering brain parenchyma by the blood-brain barriers (BBB), ~0.1–0.2% of blood circulating antibodies can cross the BBB into brain tissue in healthy rodents and humans regardless of antibody specificities [11–13]. This raises a key question as to whether chronic presence of low titers of anti-NMDAR1 autoantibodies in human blood may impair human cognitive functions and contribute to the pathogenesis of schizophrenia and other psychiatric disorders. Mouse models carrying anti-NMDAR1 autoantibodies have been generated via active immunization of NMDAR1 holoreceptor [14] or mixtures of long NMDAR1 peptides [15]; but different behavioral phenotypes were reported. Neither the human studies nor the mouse studies have so far however characterized anti-NMDAR1 autoantibodies at the level of individual antigenic epitopes. Here, we generated mice chronically carrying low titers of blood anti-NMDAR1 autoantibodies against a single NMDAR1 antigenic epitope. Since disrupted NMDAR neurotransmission causes hyperlocomotion, impaired sensorimotor gating, and deficient memory, we characterized these behavioral phenotypes in the mice carrying the anti-NMDAR1 autoantibodies.

## Materials and methods

### Mouse strain

C57BL/6J mice were purchased from Jackson Labs (Bar Harbor, ME) at 8-week-old and housed in a climate-controlled animal colony with a reversed day/night cycle. Food (Harlan Teklab, Madison, WI) and water were available *ad libitum*, except during behavioral testing. Active immunization was conducted after a week acclimation. A few microliters of blood were taken from mouse tail vein at multiple time-points (WK3, WK4, WK5, WK7, WK9, WK13, WK24) after immunization to detect production of anti-NMDAR1 autoantibodies. Behavioral testing began when mice were 24 weeks post-immunization or stated otherwise. All testing procedures were approved by UCSD or local VA Animal Care and Use Committee prior to the onset of the experiments. Mice were maintained in American Association for Accreditation of Laboratory Animal Care approved animal facilities at the local Veteran's Administration Hospital or UCSD campus. These facilities meet all Federal and State requirements for animal care.

### Active immunization

Peptides P1 and P2 from mouse NMDAR1 were synthesized by *Biomatik*. Selection of the peptides was based on immunogenicity (http://www.cbs.dtu.dk/services/NetMHCIIpan/), solubility, and surface localization on NMDAR1 proteins. The P1 (KLVQVGIYNGTHVIPNDRKI) and the P2 (TIHQEPFVYVKPTMSDGTCK) peptides are located in the amino terminal domain and the ligand binding domain of mouse NMDAR1 protein, respectively. *Mycobacterium tuberculosis*, H37 RA (Difco) and Incomplete Freund's Adjuvant (Bacto) were purchased. The peptide immunization was conducted as described for EAE mouse model [16]. In brief,

the P2 peptide was dissolved in PBS at a concentration of 4 mg/ml. An equal volume of the P2 solution is completely mixed with complete Freund's adjuvant (containing 4 mg/ml desiccated *M. Tuberculosis*, H37 RA in Incomplete Freund's Adjuvant) to generate a thick emulsion. Mice were injected with 100 ul of emulsion subcutaneously divided equally at 3 sites on mouse flank. For the injection of the control mice, the same emulsion was generated but without the P2 peptide. Blood was taken from mouse tail vein for detection of anti-NMDAR1 autoantibodies.

## Immunohistochemistry

Immunohistochemistry (IHC) was conducted as previously described [17, 18] to evaluate the production of anti-NMDAR1 autoantibodies against the P2 peptide and their semi-quantifications. Mouse blood was diluted at 1:200 with antibody diluent solution (DAKO, S080983-2) as the primary antibodies for IHC on wildtype mouse brain paraffin sections. Mouse anti-NMDAR1 monoclonal antibody (BD, cat. 556308) was diluted at 1:40,000 as a positive control. ImmPRESS peroxidase-micropolymer conjugated horse anti-mouse IgG (Vector Labs, MP-7402) was used as the secondary antibody. Chromogenic reaction was conducted with ImmPACT NovaRED Peroxidase Substrate (Vector Labs, SK-4805). Slides were mounted with Cytoseal 60 mounting medium (Richard-Allan Scientific, 8310–16). Optical intensities were measured with *Image J*, and differential intensities between hippocampal CA1 *st oriens* and *corpus callosum* were used as the surrogates for the levels of the anti-NMDAR1 autoantibodies. The anti-NMDAR1 autoantibodies persisted more than 1 year in mouse blood. Mouse brains were fixed ~10 months after immunization. Immunohistochemical analysis was conducted to investigate expression of NMDAR1, GluR1 and GFAP in the hippocampus, cortex, and striatum. Rabbit anti-NMDAR1 (Abcam, Ab17345), GluR1(Abcam, Ab31232), GFAP (Abcam, ab68428) antibodies were diluted at 1:5,000, 1:20,000, 1:10,000, respectively, for immunohistochemical staining.

## Peptide blocking experiments

Peptide blocking IHC experiments were conducted to evaluate the specificity of the staining by the autoantibodies against the NMDAR1 P2 antigenic peptide. Bovine serum albumin (BSA), the P1 peptide, the P2 peptide were pre-incubated with the diluted mouse blood containing the anti-NMDAR1 autoantibodies at the concentration of 25 ng/ul for 1 hour at room temperature, respectively. After pre-incubation, these mixtures were used as the primary antibodies for IHC staining on paraffin brain sections of wildtype mice.

## Cell-based assay

Immunofluorescence analysis of mouse anti-NMDAR1 autoantibodies were conducted using BIOCHIP (Euroimmun, FB 112d-1005-51) with a positive control of human anti-NMDAR1 autoantibody (Euroimmun, CA 112d-0101). Mouse anti-NMDAR1 monoclonal antibody (BD, cat. 556308), diluted at 1:10,000 with antibody diluent solution (DAKO, S080983-2), was also included as a positive control for the cell-based assay. Serum from mice immunized with the NMDAR1 P2 peptide was diluted at 1:10 with DAKO antibody diluent solution. Goat anti-Human IgG (H+L) Fluorescein (Vector Lab, FI-3000) and goat anti-mouse IgG (H+L) Alexa Fluor 568 (Invitrogen, A11004) were diluted at 1:1,000 as the secondary antibodies for immunofluorescence staining. The cell-based assay was conducted as recommended by the Euroimmun manufacturer. Fluorescence staining was examined using microscope EVOS FL (ThermoFisher, Scientific).

### One-Step quick assay

The NMDAR1 P2 peptide or NMDAR1 ligand binding domain was fused with GFP tagged with 6His for the One-Step quick assay to screen immunized mice for the generation of anti-NMDAR1 autoantibodies [19]. The fusion genes were synthesized and cloned into pET-21d vector and transformed into BL21 (DE3) pLysS E coli cells (EMD, 70236–3). Over-expression and purification of NMDAR1-GFP fusion proteins and subsequent One-Step assay were conducted as described [19].

### Video-Tracker

All mouse behavioral tests were conducted during their dark cycle with lights on for the entire duration of the test. Locomotor activity was measured using a Video-Tracker (VT) system in open field for 60 min as previously described [17]. Briefly, mice were first acclimated to the testing room for 60 min before placed into white plastic enclosures ($41 \times 41 \times 34$ cm$^3$) surrounded by an opaque plastic curtain. A video camera, mounted 158 cm above the enclosures, generated the signal for the Ethovision 3.0 (Noldus; Leesburg, VA) to record distance moved and time/entries in specified zones.

### Behavioral pattern monitor

The mouse Behavioral Pattern Monitor (mouse BPM) was used to record exploratory and investigatory behavior according to our previously published methods [20, 21]. Each mouse BPM chamber (San Diego Instruments, San Diego, CA) is a transparent Plexiglas box with an opaque 30 x 60 cm floor, enclosed in a ventilated isolation box. The position of the mouse in $x$, $y$ coordinates is recorded by a grid of $12 \times 24$ infrared photobeams located 1 cm above the floor. A second row of 16 photobeams (parallel to the long axis of the chamber, located 2.5 cm above the floor) is used to detect rearing behavior. Holepoking behavior is detected by 11, 1.4-cm holes that are situated in the walls (3 holes in each long wall, 2 holes in each short wall) and the floor (3 holes); each hole is equipped with an infrared photobeam. Photobeam status is sampled every 55 ms and raw beam breaks are transformed into (x, y, t, event) ASCII data files composed of the (x, y) location of the mouse in the chamber with a resolution of 1.25 cm, the duration of each event (t) and whether a holepoke or rearing occurred (event). The measures assessed were distance traveled (a measure of locomotor activity), total rearings and total hole-pokes (measures of investigatory behavior). Mice were acclimated to the testing room for 60 min prior to a 60 min session in the BPM. Testing room was dimly illuminated with a red light and chambers were kept dark.

### T-maze

Spatial working memory and/or novelty detection was tested using the T-maze. The T-maze apparatus is constructed of black Plexiglas (main stem is 45 cm long, 10 cm wide, 24 cm high; each side arm is 35 cm long, 10 cm wide, 24 cm high). Horizontal sliding doors separate the side arms from the stem. A start box, 8 cm in length, on the main stem is also separated by a horizontal sliding door. Testing was conducted under dim red light and white noise by an experimenter blind to group status. On the first day, mice were brought to the testing room for 60 min and handled for 5 min each. On the following day, mice were acclimated to the testing room for 60 min and placed in the T-maze to explore the maze for 5 min. On the third day, mice were again acclimated to the testing room for 60 min prior to spontaneous alternation testing. Each mouse was tested in a session of 8 successive free-choice trials. At the beginning of each trial, the mouse is placed in the start chamber for 30 sec. When the door is opened, the

mouse is allowed to freely explore the maze. After the mouse chooses an arm, the door is closed and the mouse is kept in the arm for 30 sec. The chosen arm and the latency to choose the arm were recorded. Subsequent trials are run in the same manner with the mouse confined to both the start chamber and the arm for 30 sec on each trial. Mice performed a total of 8 trials for a total of 7 possible alternations.

### Prepulse inhibition

Mouse startle reactivity and prepulse inhibition (PPI) were measured with startle chambers (SR-LAB, San Diego Instruments, San Diego, CA) as described [17].

### Fear conditioning

Fear conditioning was conducted as described in previous studies [22, 23] using 4 automated fear conditioning chambers (Med Associates Inc., St. Albans, Vermont). The percentage of time freezing was determined using video analysis software (Video Freeze, Med Associates Inc.). The Motion Index Threshold was set at 30 frames with a minimum freeze duration of 18 frames [24]. Footshocks were delivered through the 36 stainless steel rods located on the floor of the chamber. On Day 1 fear acquisition, after a 60-min habituation period in an adjacent room, mice were placed in the conditioning chamber. After an acclimation period (2 min), mice were presented with a tone (CS: 75 dB, 4 kHz) for 20 sec that co-terminated with a foot shock (US: 1 sec, 0.5 mA). A total of 5 tone-shock pairings were presented with an inter-trial interval of 40 sec. Freezing was measured during tone presentations and for 40 sec after shock. Mice were placed back in their home cage 40 sec after the final shock. The chambers were cleaned with water after each session. On Day 2, mice were re-exposed to the conditioning chamber to assess context memory. The habituation period and features of the chamber were identical to those used during conditioning. Mice were then placed in the chambers for 16 min and tested for freezing during which time no shocks or tones were presented and freezing was scored. On Day 3, mice were tested for cued fear and fear extinction. The context of the chambers was altered across several dimensions (tactile, odor, visual) for this test in order to minimize generalization from the conditioning context. After a 5 min acclimation period, during which time no tones were presented ("pre-tone"), 32 CS tones were presented to test cued fear and cued fear extinction. On Day 4, recall of fear extinction was examined. After a 5 min acclimation period without CS tone, 12 CS tones were presented to test recall of fear extinction.

### Statistical analysis

R programming was used for statistical analyses, effect size, and power. Analysis of variance (ANOVA) with anti-NMDAR1 autoantibody and sex as between-subjects factor, block and prepulse intensity as within-subjects factors were performed on the %PPI data and total distance traveled. For fear conditioning, ANOVA was conducted with anti-NMDAR1 autoantibody and sex as between-subjects factor and CS or time block as a within-subject factor for freezing percentages. *Post hoc* analyses were carried out using Newman-Keuls or Tukey's test. Alpha level was set to 0.05.

## Results

To generate mice carrying anti-NMDAR1 autoantibodies against a single antigenic epitope, we immunized mice with a small synthetic peptide of 20 amino acid residues that can only accommodate a single antigenic epitope for MHC class II molecules (binding antigen peptides ranging from 15–24 amino acid residues). Two NMDAR1 peptides P1 and P2, each 20 amino

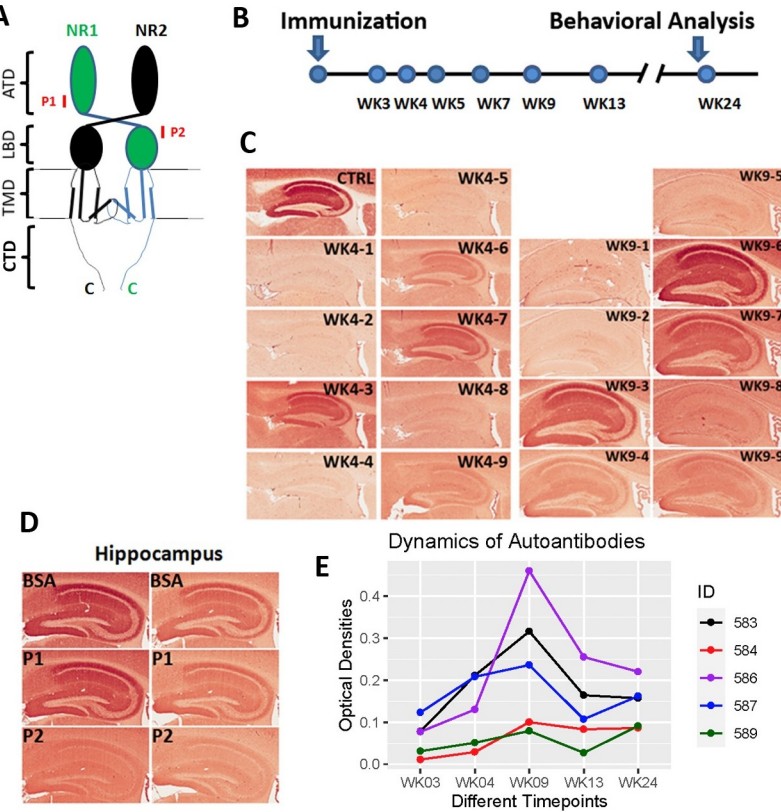

**Fig 1. Generation of anti-NMDAR1 autoantibodies in mice.** (**A**) P1 and P2 peptide antigens, each 20 amino acids long, were selected and synthesized from amino terminal domain (ATD) and ligand binding domain (LBD) of NMDAR1, respectively. TMD: transmembrane domain. CTD: C-terminal domain. (**B**) After immunization, blood was collected at different timepoints from week 3 (WK3) to week 24 (WK24). Behavioral studies were conducted after the last blood collection. (**C**) Wildtype mouse paraffin brain sections were used for detection of anti-NMDAR1 autoantibodies against the P2 antigen. Blood of immunized mice was diluted at 1:200 for the IHC analysis. CTRL: hippocampal staining with a commercial mouse anti-NMDAR1 monoclonal antibody (1:40,000 dilution) as the positive control. At week 4 (WK4), #3, #6, #7, and #9 mice were positives for anti-NMDAR1 autoantibodies. At week 9 (WK9), #3, #4, #6, #7, and #9 were positives. (**D**) Pre-incubation of the P2 antigen with positive mouse sera blocked the staining of mouse hippocampus, whereas pre-incubation of either the P1 antigen or BSA had no blocking effects on the staining. (**E**) Differences of optical intensities between hippocampal CA1 st oriens and corpus callosum were quantified with Image J and served as surrogates for the levels of anti-NMDAR1 autoantibodies in different mice from week 3 to week 24.

acids long, were selected from the amino terminal domain (ATD) and the ligand binding domain (LBD) of NMDAR1, respectively, and were synthesized by *Biomatik* (Fig 1A). The P2 peptide was emulsified with Complete Freund's Adjuvant (CFA) to immunize 2-month-old wildtype C57BL/6J male mice. To generate low titers of anti-NMDAR1 autoantibodies, mice received only a dose of primary immunization without a booster. Blood was collected via mouse tail vein at different time-points after the immunization (Fig 1B). Wildtype mouse paraffin brain sections were used for detection of anti-NMDAR1 autoantibodies by immunohistochemistry (IHC). Anti-NMDAR1 autoantibodies were detectable by IHC 3 weeks after immunization, and 5 out of 9 immunized mice developed the anti-NMDAR1 autoantibodies in blood by week 9 (WK9) (Fig 1C). To confirm the specificity of the anti-NMDAR1 autoantibodies, we conducted peptide blocking in IHC. As expected, binding of the anti-NMDAR1 autoantibodies to hippocampal NMDAR1 was blocked by the P2 peptide, but not by either the

P1 peptide or bovine serum albumin (BSA) (Fig 1D). Optical intensities of CA1 staining (CA1 *st oriens*), after subtracting the background intensity from *corpus callosum*, were used as surrogates for the levels of the anti-NMDAR1 autoantibodies from WK3 to WK24. The levels of the anti-NMDAR1 autoantibodies varied a few folds between different mice at WK3 and were gradually stabilized by WK24 (Fig 1E). The titers of the anti-NMDAR1 autoantibodies in our mice (1 to 200 dilution in IHC) are 1–3 orders of magnitude lower than the titers of anti-NMDAR1 autoantibodies in the blood of patients with anti-NMDAR1 encephalitis (1 to 2,000–500,000 dilution in IHC) [25]. In this small male cohort, 5 males developed the anti-NMDAR1 autoantibodies, whereas 7 males did not have the anti-NMDAR1 autoantibodies (including 3 control males immunized with the CFA only).

To examine whether the anti-NMDAR1 autoantibodies against the P2 peptide can recognize NMDAR1 proteins in their native conformation, we conducted cell-based assays (Euroimmun) where NMDAR1 proteins were expressed on transfected HEK293 cells. Mouse serum containing the anti-NMDAR1 autoantibodies against the P2 peptide recognizes the NMDAR1 proteins on the transfected HEK293 cells (Fig 2A). As expected, immunohistochemical analysis using both human anti-NMDAR1 autoantibodies and mouse anti-NMDAR1 P2 autoantibodies demonstrated a complete co-localization of their staining (Fig 2B). After a series of dilutions of mouse serum, we found that titers of the anti-NMDAR1 autoantibodies against the P2 peptide range from 1:10–1:100 using cell-based assays among individual mice. These titers are comparable for low titers of blood anti-NMDAR1 autoantibodies in the general human population and psychiatric patients [8, 26], supporting the relevance of our mouse modeling for chronic effects of low titers of blood anti-NMDAR1 autoantibodies on human mental health.

All of the mice carrying the anti-NMDAR1 autoantibodies were healthy. We examined several behaviors modulated by NMDAR in mice 24 weeks after immunization to investigate chronic effects of low titers of the anti-NMDAR1 autoantibodies. We found no differences in total distance traveled between the negative control mice and the mice carrying the anti-NMDAR1 autoantibodies against the P2 peptide (Fig 3A). No difference was observed in either center duration or center frequency (Fig 3B and 3C) between the two groups. We next examined sensorimotor gating as measured by prepulse inhibition between the two groups. There was no difference in either startle habituation (Fig 3D) or prepulse inhibition of startle (Fig 3E). Mouse cognitive function on spatial working memory and/or novelty detection was assessed with T-maze spontaneous alternation. A trend of impaired spontaneous alternation (p = 0.06) was found in mice carrying the anti-NMDAR1 autoantibodies (Fig 3F). Mice were further tested with fear conditioning to measure fear learning and memory as well as fear extinction. Mice carrying the anti-NMDAR1 autoantibodies appeared to acquire stronger fear memory by the end of acquisition in either post-shock (Fig 4A) or during the tone (CS) (Fig 4B). No difference was observed in contextual fear memory between the two groups (Fig 4C). However, mice carrying the anti-NMDAR1 autoantibodies had impaired cued fear extinction learning (Fig 4D) and recall of fear extinction (Fig 4E). The raw behavior data of the pilot cohort can be found in S1 Table.

To enlarge the sample size and include female mice, we purchased a large cohort of 40 2-month-old C57BL/6J mice (20 males, 20 females) for active immunization. To improve the success rate of production of the anti-NMDAR1 autoantibodies, we used Loss-of-Resistance (LOS) syringes (BD Epilor™ Syringe, 405291) to generate more viscous antigen/CFA emulsion. Half of mice (10 males, 10 females) were immunized with the P2 peptide emulsified with the CFA; whereas the other half (10 males, 10 females) were immunized with the CFA only to serve as the control group. A month later, anti-NMDAR1 autoantibodies were examined in mouse blood. All of the 20 mice immunized with the P2 peptide generated the anti-NMDAR1

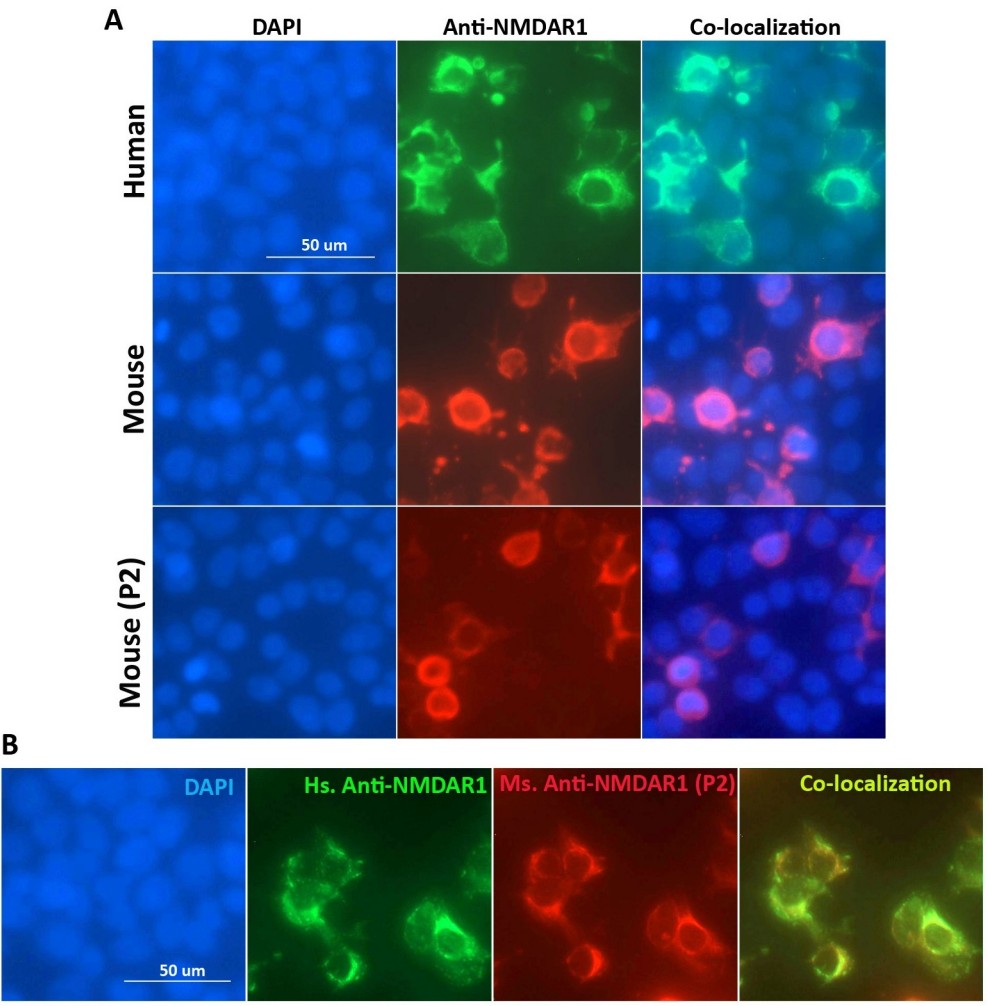

**Fig 2. Characterization of the anti-NMDAR1 autoantibodies using cell-based assays. (A)** Human NMDAR1 proteins were expressed on HEK293 cells on BIOCHIPs purchased from Euroimmun. Both anti-Human NMDAR1 autoantibody (Euroimmun) and mouse anti-NMDAR1 antibody (BD, diluted at 1:10,000) recognize the NMDAR1 proteins on HEK293 cells in cell-based assays. The anti-NMDAR1 autoantibodies (diluted at 1:10) against the P2 peptide antigens from our mice also recognize the native conformation of the NMDAR1 proteins in cell-based assays. **(B)** Co-localization of the staining between the anti-Human NMDAR1 and mouse anti-NMDAR1 P2 autoantibodies in cell-based assays. Bar: 50 μm.

autoantibodies (Fig 5), whereas none of the control mice produced the anti-NMDAR1 autoantibodies. Because of using the LOS syringes for emulsion, the success rate of the anti-NMDAR1 autoantibodies improved from ~50% in the previous small cohort to 100% in this large cohort. All of the mice carrying the anti-NMDAR1 autoantibodies were healthy except for a control female mouse that died before behavioral analysis. Behavioral analysis was conducted 24 weeks after immunization to match the timeline of the previous small cohort (Fig 1B). A significant reduction of spontaneous alternation in T-maze was observed in mice carrying the anti-NMDAR1 autoantibodies (Fig 6A). After separating the sexes, we found that both the female and the male mice carrying the autoantibodies showed significant cognitive deficits in T-maze (Fig 6B), replicating the finding of the previous small male cohort. Statistical analyses, effect size (d = 1.296), and power (0.99) are summarized in Fig 6C. Mouse locomotor activities were

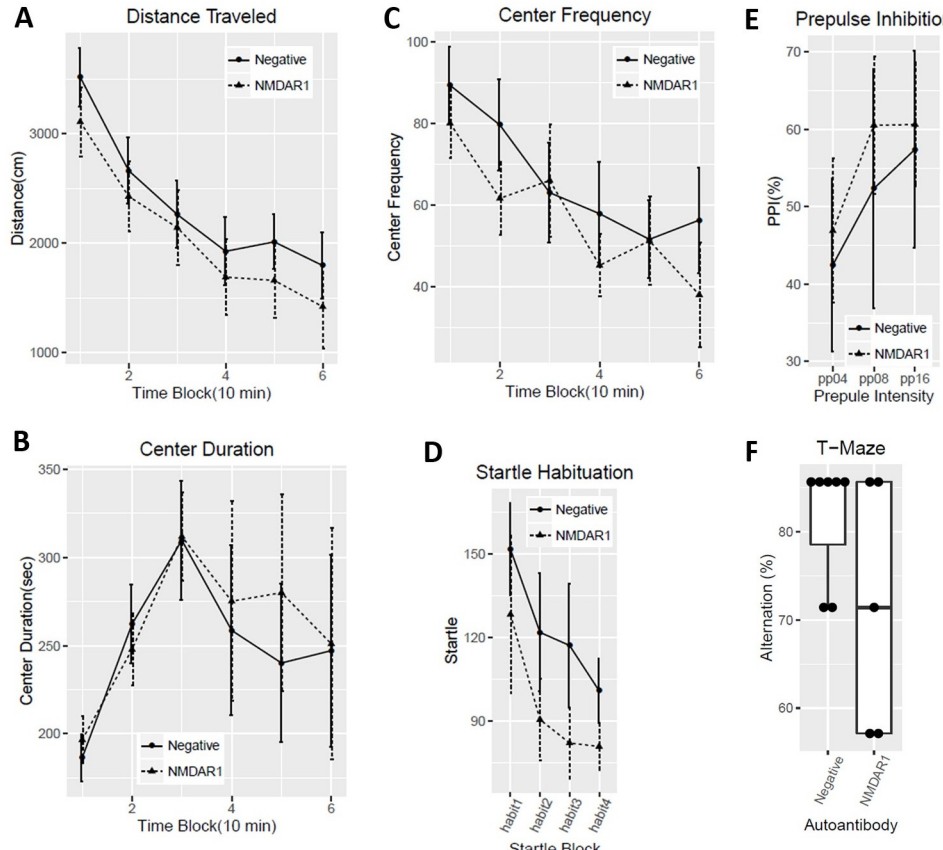

**Fig 3. Behavioral analysis of the mice in open field locomotion, prepulse inhibition, and T-maze tests.** Negative control male mice (n = 7); positive NMDAR1 male mice (n = 5). **(A)** No difference ($F_{(1,10)}$ = 0.45, p = 0.52) in total distances traveled was observed between the two groups, 10 min for each time-block. **(B)** There is no difference ($F_{(1,10)}$ = 0.06, p = 0.81) in center duration between the two groups. **(C)** No difference ($F_{(1,10)}$ = 0.38, p = 0.55) was observed between the two groups in frequencies for mice entering the center. **(D)** There is no difference ($F_{(1,10)}$ = 0.07, p = 0.8) in startle habituation between the two groups of mice. **(E)** No difference ($F_{(1,10)}$ = 0.05, p = 0.84) was observed in prepulse inhibition across pp4, pp8, pp16 prepulse levels between the two groups. **(F)** Alternation percentages of individual mice were present as boxplot. There was a trend (t(10) = 1.81, p = 0.06, one-tail *t* test) of lower spontaneous alternation in T-maze in mice carrying the anti-NMDAR1 autoantibodies. All data are present as Mean+SEM.

examined using the Video-Tracker used in the small cohort. No difference was observed in total distance travelled (Fig 6D). There was a sex effect, but not the autoantibody effect, in center duration and center frequency (Fig 6E and 6F). No difference was found in either startle habituation (Fig 6G) or prepulse inhibition (Fig 6H). Mice were further tested to confirm the fear conditioning phenotypes from the previous pilot male cohort. We did not observe differences in fear acquisition in either post-shock (Fig 7A) or during CS (Fig 7B). There was no autoantibody effect on contextual fear memory, except for a sex effect (Fig 7C). We detected a sex effect in fear extinction learning, but not an autoantibody effect (Fig 7D). No effect of the autoantibody was observed in recall of fear extinction (Fig 7E). The large cohort of mice replicated the T-maze deficit, but not the fear conditioning phenotypes, from the pilot cohort of male mice. The raw behavior data of the large cohort can be found in S2 Table.

Since behavioral abnormalities are particularly susceptible for variations, we consider that another large cohort of mice is necessary for replication to conclude the effects of the anti-

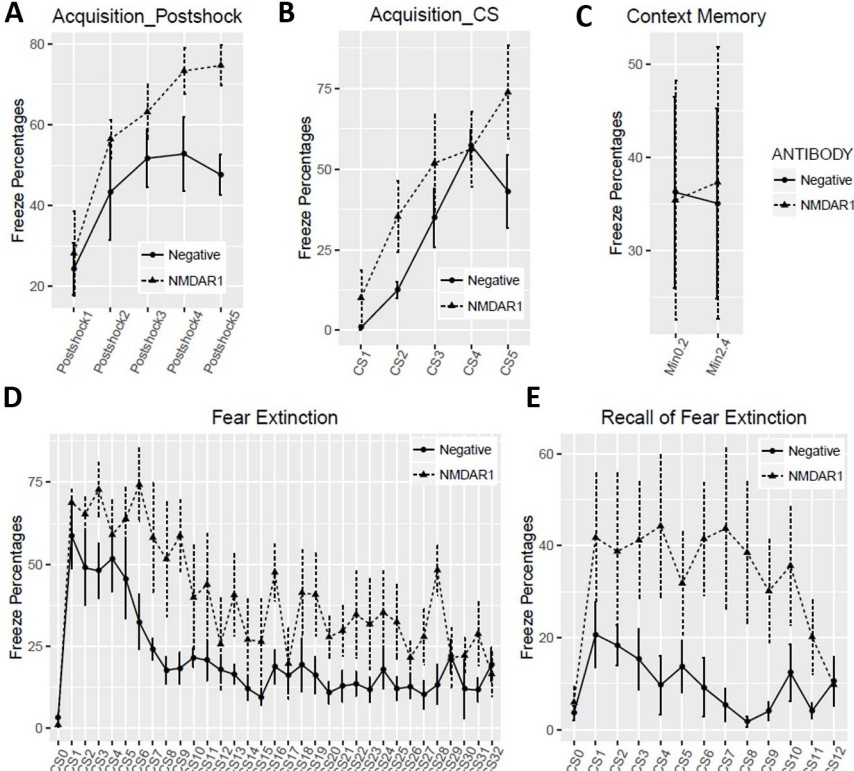

**Fig 4. Fear conditioning 4-day fear conditioning tests were conducted as previously described [27].** On Day 1 fear acquisition, mice carrying the anti-NMDAR1 autoantibodies displayed higher freezing percentages after shock (**A**, F (1,10) = 7.06, p = 0.024) or during the tone (**B**, F(1,10) = 7.03, p = 0.024). (**C**) On Day 2, contextual fear memory was assessed. Freezing percentages were calculated for the first two blocks, 2 min for each block. There was no difference (F (1,10) = 0.002, p = 0.97) between the two groups. (**D**) On Day 3, fear extinction was assessed with 32 tones (CS) without foot shock. Slower fear extinction (F91,10) = 19.35, p = 0.001) was observed in mice carrying the anti-NMDAR1 autoantibodies. CS0: pre-tone. (**E**) On Day 4, recall of fear extinction was examined with 12 CS without foot shock. An impaired recall of fear extinction (F(1,10) = 7.9, p = 0.019) was detected in mice carrying the anti-NMDAR1 autoantibodies. CS0: pre-tone. All data are present as Mean+SEM.

NMDAR1 autoantibodies on mouse behavioral phenotypes. We purchased the second large cohort of 40 2-month-old C57BL/6J mice (20 males, 20 females) for active immunization. The immunization was conducted exactly as described in the previous large cohort. A month later, anti-NMDAR1 autoantibodies were examined in the blood of all 40 immunized mice. To reduce the use of research animals, we developed an *in vitro* One-Step quick assay for a rapid screening of the presence of anti-NMDAR1 autoantibodies (Fig 8A and 8B). All of the 20 mice immunized with the P2 peptide generated the anti-NMDAR1 autoantibodies by the One-Step assay, whereas none of the control 20 mice immunized with CFA produced the anti-NMDAR1 autoantibodies. The One-Step assay showed 100% consistence with immunohisto-chemical staining, providing an excellent alternative method for antibody screening without using mouse brain sections.

The same previous behavioral test battery was used for this large replication cohort of mice exception for two modifications. First, mouse locomotion was tested using Behavioral Pattern Monitor that is similar to the Video-Track but can additionally examine pokes and rears, the exploratory behaviors. Second, we shortened the interval between the immunization and the behavioral tests from 24 weeks to 11 weeks to examine time-dependent effects of the

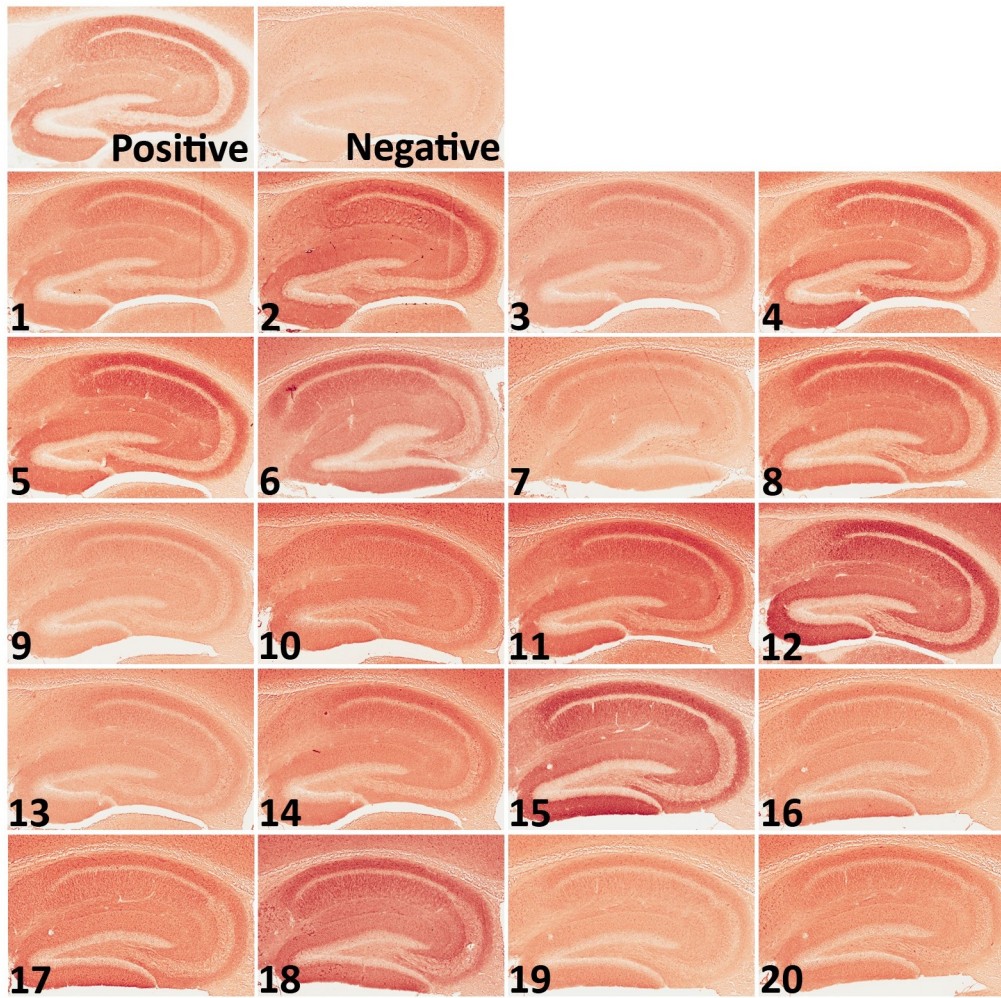

**Fig 5. Generation of the anti-NMDAR1 autoantibodies in a large cohort.** A large cohort of 40 mice (20 males and 20 females) was immunized with either the P2 plus the CFA or the CFA only. After improving the antigen emulsification using the LOS syringes, all mice (10 males and 10 females) immunized with the P2 plus the CFA generated anti-NMDAR1 autoantibodies. No difference was observed between the males and females. All of the control mice (10 males and 10 females) immunized with the CFA only are negative for the autoantibodies against the NMDAR1 P2 antigen.

autoantibody on mouse cognitive functions. Defective spontaneous alternation in T-maze was validated again in mice carrying the anti-NMDAR1 autoantibodies with a *p* value reaching 2.02E-08 (Fig 9A). Such a robust behavioral deficit can almost be used to differentiate which individual mice carrying the autoantibodies except for 5 mice overlapping with the control group. As expected, there is no sex effect, both female and male mice carrying the autoantibodies showed impaired spontaneous alternation in T-maze (Fig 9B). Statistical analyses, effect size (d = 2.31), and power (1) are summarized in Fig 9C. Mouse locomotion and exploratory activities were investigated using Behavioral Pattern Monitor (BPM). There was no difference between the two groups of mice in total distance travelled (Fig 9D). There was no effect of the autoantibodies on center duration (Fig 9E), center frequency (Fig 9F), pokes (Fig 9G), and rears (Fig 9H). No difference was found in either startle habituation (Fig 9I) or prepulse inhibition (Fig 9J). Mice were further tested with fear conditioning. We did not observe differences

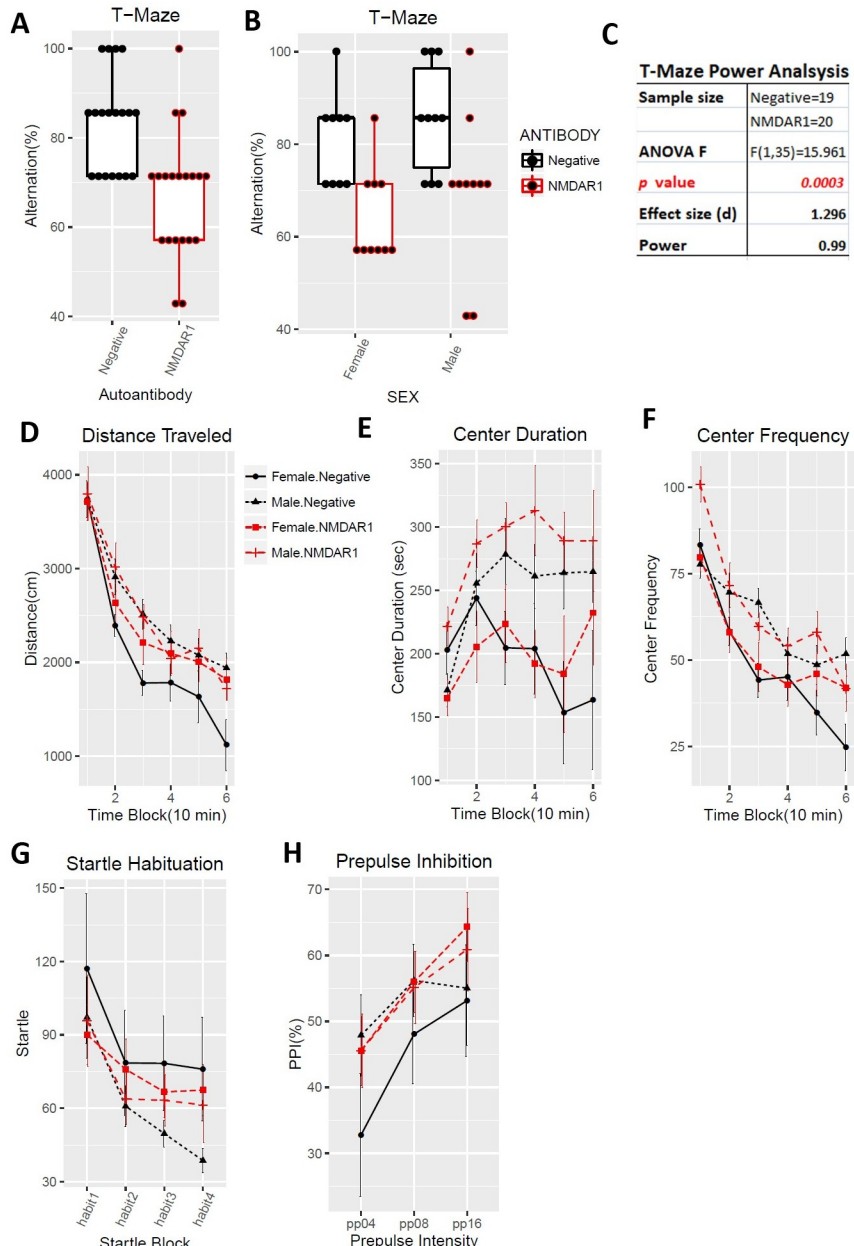

**Fig 6. Impaired spontaneous alternation in T-maze.** In the large cohort of 40 immunized mice, one female control mice immunized with the CFA only died before behavioral analysis. All other mice are healthy. **(A)** Alternation percentages of individual mice were present as boxplots. A significant reduction of spontaneous alternation was replicated in the large cohort (F(1,35) = 15.96, p = 0.0003). **(B)** Both male and female mice carrying the anti-NMDAR1 autoantibodies displayed impaired spontaneous alternation, replicating the T-maze performance of the previous small male cohort. **(C)** This large cohort of mice provided a sufficient statistical power (0.99) to analyze the effect size (d = 1.296) of the anti-NMDAR1 autoantibodies between the mice with or without the anti-NMDAR1 autoantibodies. **(D)** There is neither sex (F(1,31) = 2.18, p = 0.15) nor autoantibody (F(1,31) = 0.7, p = 0.41) effect on total distance travelled in the open field. **(E)** Male mice spent significant more time (F(1,31) = 10.37, p = 0.003) in the center than female mice, but there is no autoantibody effect (F(1,31) = 1.05, p = 0.31). **(F)** Consistent with the center time, a significant more center frequency (F(1,31) = 6.38, p = 0.017) was observed in male mice. There is no effect of the autoantibody (F(1,31) = 0.99, p = 0.33). **(G)** There is neither sex (F(1,35) = 1.11, p = 0.3) nor autoantibody (F(1,35) = 0.01, p = 0.93) effect on startle habituation. No interactions between startle block and the autoantibody (F(3,105) = 1.57, p = 0.2). **(H)** Neither sex (F(1,35) = 0.3, p = 0.57) nor the autoantibody (F(1,35) = 0.93, p = 0.34) effect was observed in PPI. All data are present as Mean+SEM.

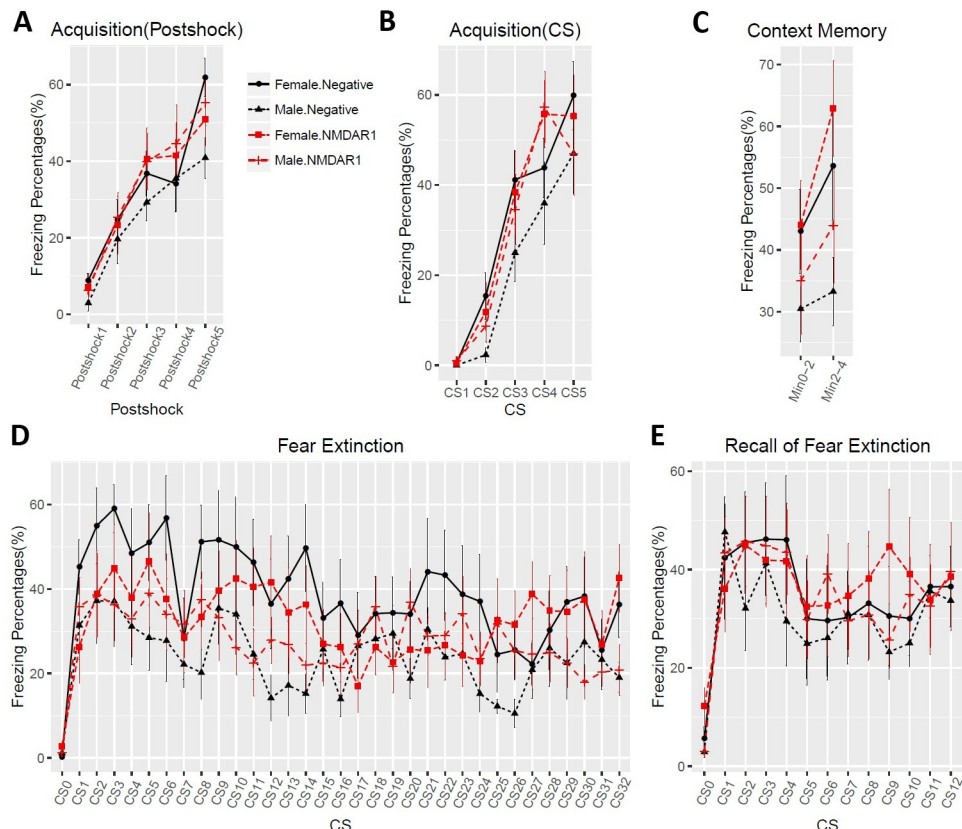

**Fig 7. No effects of the anti-NMDAR1 autoantibodies on fear conditioning. (A)** In Day1 fear acquisition, there is neither sex ($F_{(1,31)} = 0.35$, p = 0.56) nor autoantibody ($F_{(1,31)} = 0.51$, p = 0.48) effect on post-shock freezing time. **(B)** Neither sex ($F_{(1,31)} = 1.93$, p = 0.17) nor autoantibody ($F_{(1,31)} = 0.61$, p = 0.44) effect was observed during CS tone. **(C)** In Day2 context memory, female mice have significant higher freezing percentages ($F_{(1,35)} = 5.2$, p = 0.029) than the male mice, but there is no autoantibody effect ($F_{(1,35)} = 0.9$, p = 0.35). **(D)** In Day3 fear extinction, there is a trend of sex effect ($F_{(1,35)} = 2.94$, p = 0.095), but without autoantibody effect ($F_{(1,35)} = 0.04$, p = 0.85). CS0: pre-tone. **(E)** In Day4 recall of fear extinction, neither sex ($F_{(1,35)} = 0.24$, p = 0.63) nor autoantibody ($F_{(1,35)} = 0.24$, p = 0.63) effect was observed. CS0: pre-tone. All data are present as Mean+SEM.

in fear acquisition in either post-shock (Fig 10A) or during CS (Fig 10B). There was no autoantibody effect on contextual fear memory (Fig 10C). We did not detect effect of the autoantibodies on fear extinction (Fig 10D). There was, however, a significant interaction ($F_{(12,420)} = 2.96$, p = 0.00056) between the autoantibodies and the CS in recall of fear extinction (Fig 10E). The raw behavior data of the replication cohort can be found in S3 Table.

Behavioral phenotypes from all 3 independent cohorts of mice are summarized and compared in Table 1. Deficient spatial working memory and/or novelty detection in T-maze was consistently observed across all 3 cohorts of mice carrying the anti-NMDAR1 autoantibodies. No differences in locomotion, exploration, startle, and prepulse inhibition were found between the control mice and mice carrying the autoantibodies. Fear conditioning phenotypes were not consistently detected in different cohorts of mice, suggesting that these phenotypes may be either mild or false positives.

Anti-NMDAR1 autoantibodies against the P2 antigen persisted in mouse blood for more than 14 months. Since chronic presence of the circulating anti-NMDAR1 autoantibodies impairs mouse spatial working memory and/or novelty detection in T-maze, we examined the

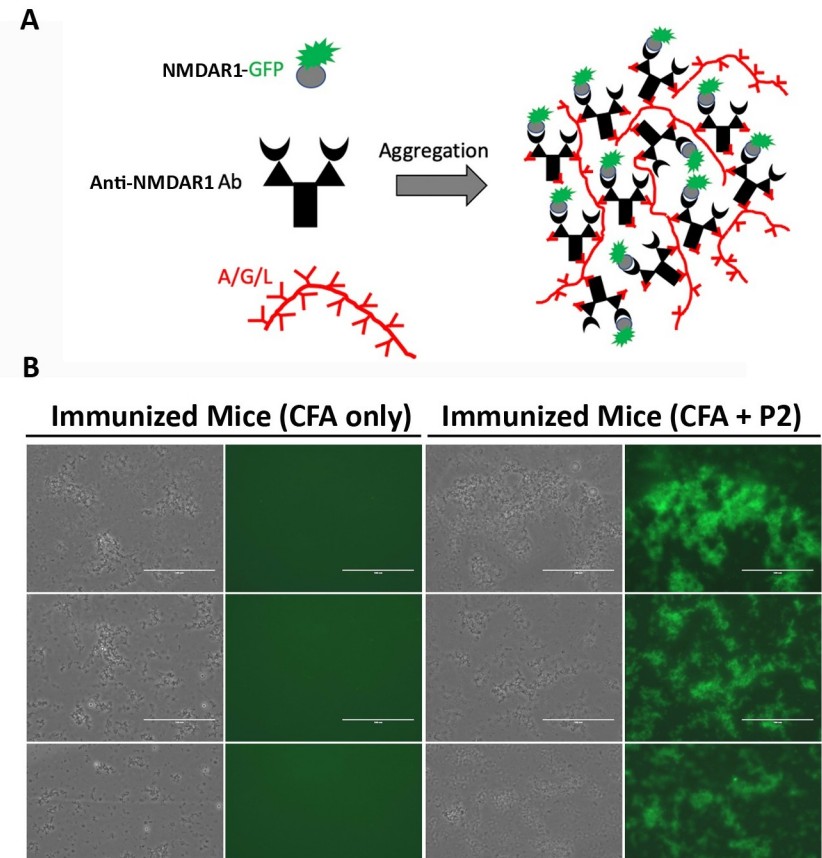

**Fig 8. Generation of the anti-NMDAR1 autoantibodies in a large replication cohort.** Another large cohort of 40 mice (20 males and 20 females) was immunized with either the P2 plus the CFA or the CFA only. **(A)** A new One-Step quick assay was developed to screening the presence of the anti-NMDAR1 P2 autoantibodies in the blood of all 40 immunized mice. Either the P2 peptide or the LBD of NMDAR1 was fused with GFP. The fusion proteins were purified from E coli for the assay. The protein A/G/L instantly aggregates antibody-antigen-GFP complexes that emit strong green fluorescence [19]. **(B)** All of the 20 mice immunized with the P2 peptide antigens developed anti-NMDAR1 autoantibodies one month after immunization, whereas all of the control 20 mice immunized with CFA only were negative. Bar: 100 μm.

brains of 5 pairs of mice for reduction of NMDAR1 proteins in different brain regions, particularly hippocampus. GFAP staining for astrogliosis and expression of GluR1 proteins were also examined in mouse hippocampus. We did not observe significant differences in the levels of NMDAR1, GluR1, and GFAP in hippocampus, cortex, and striatum between the two groups of mice (Fig 11).

## Discussion

Our studies demonstrated that chronic presence of low titers of blood circulating anti-NMDAR1 autoantibodies against the P2 antigenic epitope is sufficient to impair mouse spatial

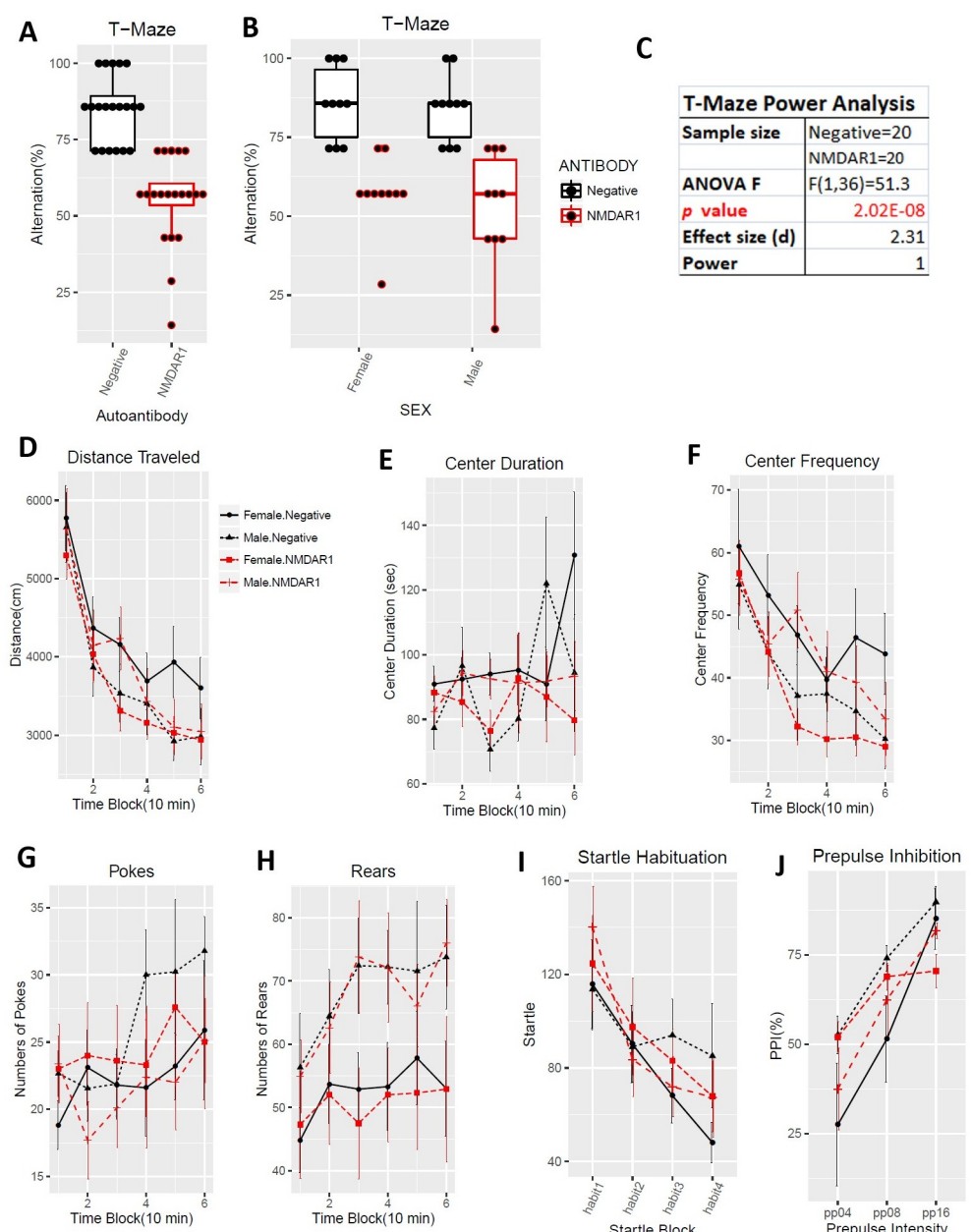

**Fig 9. Replication of deficient spontaneous alternation in T-maze.** All mice are healthy. **(A)** Alternation percentages of individual mice were present as boxplots. Spontaneous alternation was dramatically impaired in mice carrying the anti-NMDAR1 autoantibodies (F(1,36) = 51.3, p = 2.02E-08), replicating our finding across all 3 different mouse cohorts. **(B)** Both male and female mice carrying the anti-NMDAR1 autoantibodies displayed impaired spontaneous alternation, replicating the T-maze performance of the previous cohorts. **(C)** This large replication cohort of mice provided a sufficient statistical power (1) to analyze the effect size (d = 2.31) of the anti-NMDAR1 autoantibodies between the mice with or without the anti-NMDAR1 autoantibodies. Mouse locomotion and exploratory activity were examined using Behavioral Pattern Monitor. One control mice died during the behavioral tests. **(D)** There is neither sex (F(1,35) = 0.092, p = 0.76) nor autoantibody (F(1,35) = 0.44, p = 0.51) effect on total distance travelled in the open field. **(E)** Neither sex effect (F(1,35) = 0.054, p = 0.82) nor the autoantibody effect (F(1,35) = 1.33, p = 0.26) was found for center duration between the group of mice. **(F)** There is neither sex (F(1,35) = 0.021, p = 0.88) nor autoantibody (F(1,35) = 0.58, p = 0.45) effect on center frequency. **(G)** Neither sex (F(1,35) = 0.05, p = 0.83) nor autoantibody (F(1,35) = 0.23, p = 0.63) effect on pokes in Behavioral Pattern Monitor test. **(H)** There is a sex effect (F(1,35) = 5.4, p = 0.025), but no autoantibody effect (F(1,35) = 0.04, p = 0.84) on rears. **(I)** There is no sex (F(1,35) = 0.17, p = 0.68) or autoantibody (F(1,35) = 0.09, p = 0.77) effect on startle habituation. No interactions between startle block and the autoantibody (F(3,105) = 1.1, p = 0.35) was observed. **(J)** Neither sex (F(1,35) = 1.45, p = 0.24) nor autoantibody (F(1,35) = 0.03, p = 0.87) effect was observed in PPI. All data are present as Mean+SEM.

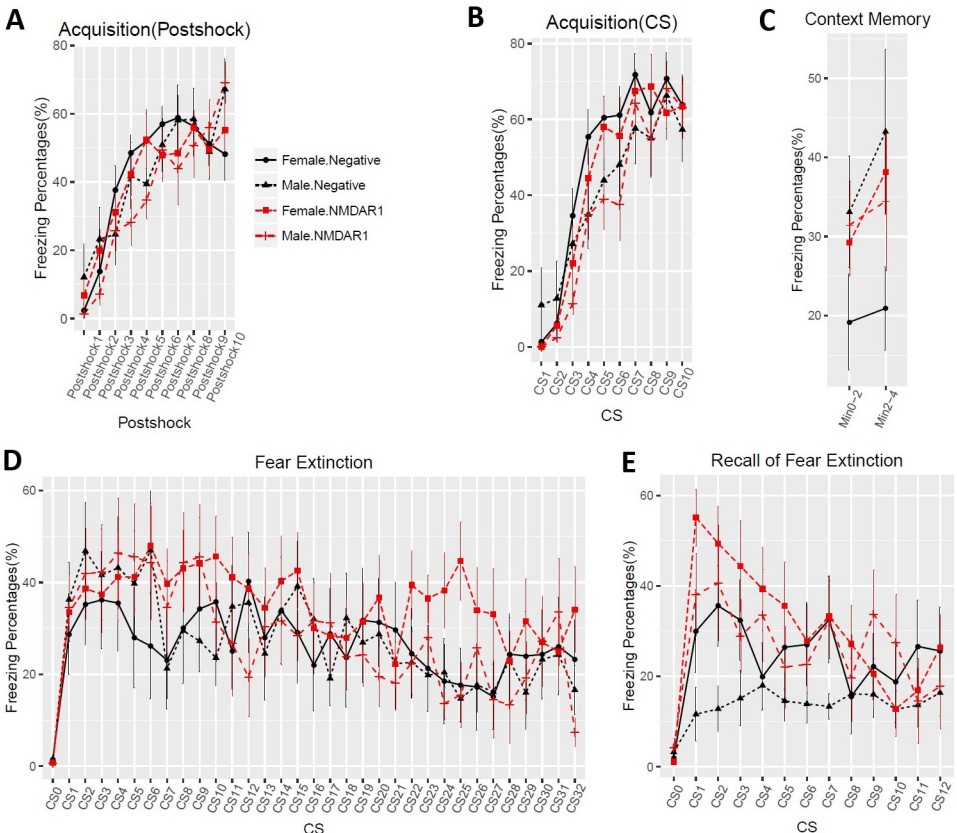

**Fig 10. Effects of the anti-NMDAR1 autoantibodies on fear conditioning in the large replication cohort. (A)** In Day 1 fear acquisition, there is neither sex ($F_{(1,36)} = 0.16$, $p = 0.69$) nor autoantibody ($F_{(1,36)} = 0.45$, $p = 0.51$) effect on post-shock freezing time. **(B)** Neither sex ($F_{(1,36)} = 1.79$, $p = 0.19$) nor autoantibody ($F_{(1,36)} = 0.53$, $p = 0.47$) effect was observed during CS tone. **(C)** In Day 2 context memory, neither sex ($F_{(1,35)} = 2.05$, $p = 0.16$) nor autoantibody ($F_{(1,35)} = 0.55$, $p = 0.46$) effect was observed. **(D)** In Day 3 fear extinction, there is no sex effect ($F_{(1,35)} = 0.25$, $p = 0.62$) or autoantibody effect ($F_{(1,35)} = 0.52$, $p = 0.48$). CS0: pre-tone. **(E)** In Day 4 recall of fear extinction, neither sex ($F_{(1,35)} = 1.26$, $p = 0.27$) nor autoantibody ($F_{(1,35)} = 2$, $p = 0.17$) effect was observed. However, there is a significant interaction ($F_{(12,420)} = 2.96$, $p = 0.00056$) between the autoantibody and CS. CS0: pre-tone. All data are present as Mean+SEM.

working memory and/or novelty detection in the integrity of blood-brain barriers. To our knowledge, our studies are the first characterization of mice carrying anti-NMDAR1 autoantibodies against a single specific antigenic epitope. It is essential to investigate functions of anti-NMDAR1 autoantibodies at the level of individual antigenic epitopes, since anti-NMDAR1 autoantibodies against different antigenic epitopes may alter NMDAR functions differently. For example, anti-NMDAR2A autoantibodies binding a specific antigenic epitope can function as an agonist [28]. It will not be surprising if anti-NMDAR1 autoantibodies against certain antigenic epitopes may act as agonists rather than antagonists. Investigation of individual antigenic epitopes across the whole extracellular domain of NMDAR1 protein will help understand the effects of different anti-NMDAR1 autoantibodies on mouse behavioral phenotypes in future studies.

The mice carrying the anti-NMDAR1 autoantibodies against the P2 antigenic epitope were healthy and exhibited no abnormalities in a number of NMDAR-modulated behaviors except for robust deficient spatial working memory and/or novelty detection. Interestingly,

**Table 1. Summary of behavioral analysis.**

|  | Pilot Cohort | Large Cohort | Replication |
|---|---|---|---|
| Mice | C: 7 (M) | C: 10 (M), 9 (F) | C: 10 (M), 10 (F) |
|  | P: 5 (M) | P: 10 (M), 10 (F) | P: 10 (M), 10 (F) |
| **T-Maze** |  |  |  |
| Spontaneous Alternation | **p = 0.06** | **p = 0.0003** *** | **p = 2.02e-08** *** |
| **Locomotion** |  |  |  |
| Total Distance | ns | ns | ns |
| Center Duration | ns | ns | ns |
| Center Frequency | ns | ns | ns |
| Pokes |  |  | ns |
| Rears |  |  | ns |
| **PPI** |  |  |  |
| Startle | ns | ns | ns |
| Prepulse Inhibition | ns | ns | ns |
| **Fear Conditioning** |  |  |  |
| Fear Acquisition | p = 0.024 | ns | ns |
| Context Fear | ns | ns | ns |
| Fear Extinction | p = 0.001 | ns | ns |
| Fear Extinction Recall | p = 0.019 | ns | p = 0.00056 # |

# Antibody X CS Interaction.

intracerebroventricular injection of cerebrospinal fluid (CSF) from anti-NMDAR1 encephalitis patients causes robust impairment of mouse novelty detection without affecting locomotor activities and other behaviors [10]. Effects of anti-NMDAR1 autoantibodies on mouse behaviors in our studies appear more restricted than either administration of NMDAR antagonists or NMDAR1 hypomorphic mice that display hyperlocomotion, deficient prepulse inhibition, and impaired context memory [29–31]. It will be interesting to know whether novelty detection and spatial working memory are more sensitive to NMDAR dysfunction than other NMDAR-modulated behaviors and cognition. Hippocampus plays a central role in novelty detection and spatial working memory. We examined expression of NMDAR1, GluR1, and GFAP in hippocampus using immunohistochemistry but did not find alteration of these proteins in mice carrying the anti-NMDAR1 autoantibodies. Our immunohistochemical analysis however may be not sensitive enough to identify subtle changes. It cannot be ruled out either that the blood circulating anti-NMDAR1 autoantibodies may preferentially infiltrate to localized brain regions to reduce NMDAR1 proteins that were not sampled by our brain sections.

Low titers of blood anti-NMDAR1 autoantibodies were found using cell-based assays in ~10% of the general human population and psychiatric patients. By using the same cell-based assays, we found that the titers of blood anti-NMDAR1 autoantibodies against the P2 antigen in our mice are low and comparable to those in healthy humans. However, most anti-NMDAR1 autoantibodies are IgM and IgA isotypes in the blood of the general human population [8, 26], whereas anti-NMDAR1 IgG and IgM autoantibodies are produced in our immunized mice. Therefore, it is important to investigate whether IgG or IgM or both anti-NMDAR1 autoantibodies are responsible for the deficient spatial working memory and/or novelty detection in mice immunized with the NMDAR1 P2 peptide antigens.

Behavioral phenotypes are particularly susceptible for non-replicability [32]. Differences in fear acquisition observed in the pilot cohort (n1 = 7, n2 = 5) cannot be replicated by either of

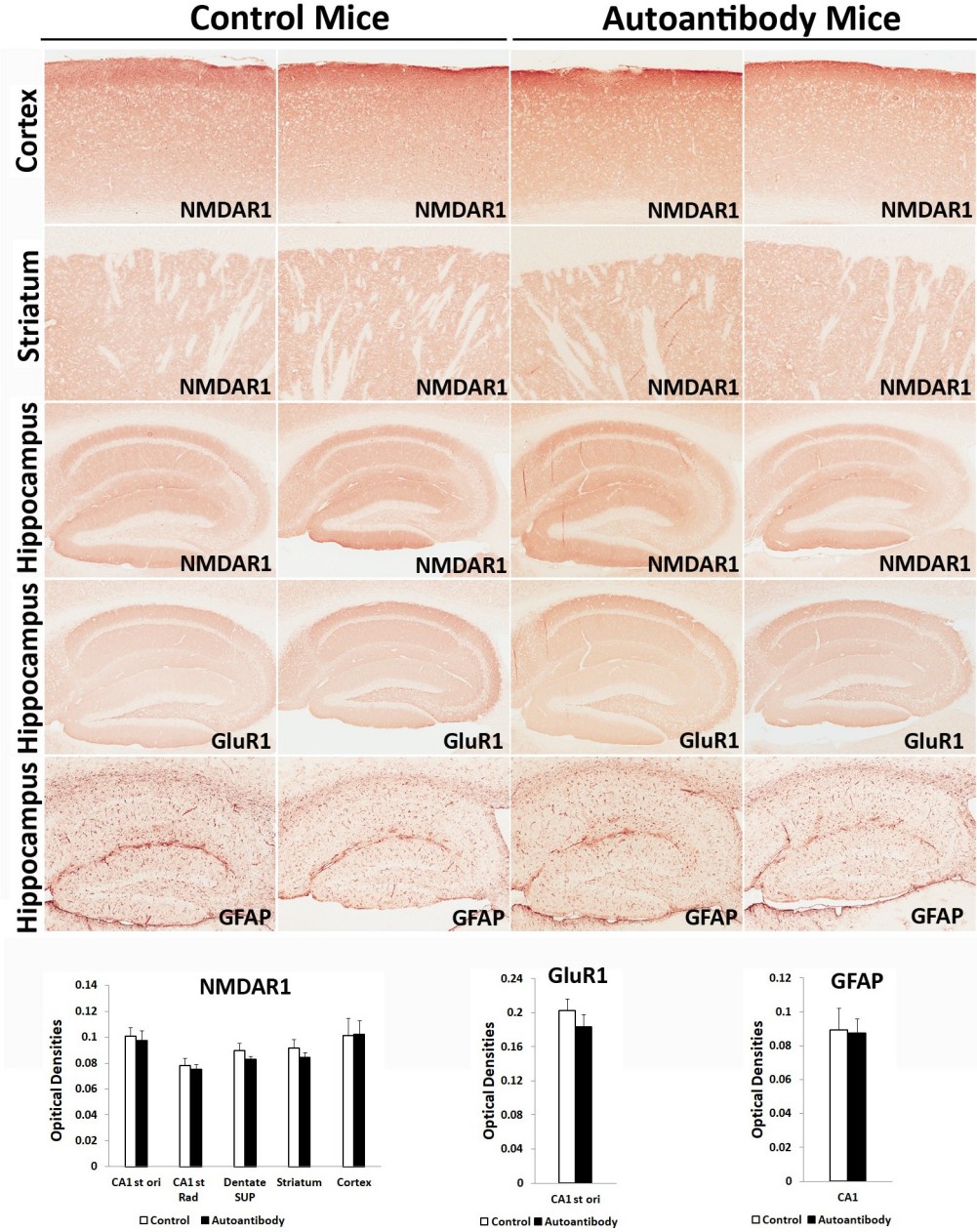

**Fig 11. Lack of significant differences of NMDAR1, GluR1, and GFAP in the brains of mice chronically carrying the anti-NMDAR1 autoantibodies.** 5 pairs of mice with or without the anti-NMDAR1 autoantibodies were sacrificed to examine expression of NMDAR1, GluR1, and GFAP in hippocampus, cortex, and striatum. Representative staining of two mice from each group are shown. Image J was used to quantify the optical densities of the staining at different brain regions. There are no significant differences in expression of NMDAR1, GluR1, and GFAP between the control mice and the mice carrying the anti-NMDAR1 autoantibodies. Data were presented as mean+SEM.

the two subsequent large cohorts of mice. Even between the two large cohorts of mice (each with 40 mice), sex effects on center duration, center frequency, and fear conditioning are not replicated with each other. Such non-replicability was also observed in recall of fear extinction between the two large cohorts. However, deficient spontaneous alternation in T-maze, a true

behavioral phenotype by the autoantibodies, is replicated very well across all 3 different mouse cohorts, suggesting that more replications with large cohort sizes are necessary to differentiate potentially false positive behavioral phenotypes.

Anti-NMDAR1 autoantibodies have been found in patients with many diseases. ~16% to 60% of patients with Alzheimer's Diseases or other dementias have anti-NMDAR1 autoantibodies in their blood, but not in CSF [33]. In addition to ovarian teratomas where anti-NMDAR1 encephalitis was originally identified, ~20% of patients with a variety of other cancers develop anti-NMDAR1 autoantibodies in their blood, but not in CSF [34]. In the general human population, prevalence of anti-NMDAR1 autoantibodies increases in blood during aging [15]. Our studies suggest that chronic presence of blood circulating anti-NMDAR1 autoantibodies may be sufficient to cause cognitive impairments in both healthy persons and patients with a variety of diseases including psychiatric disorders.

## Supporting information

**S1 Table. Raw behavioral data of the pilot cohort.** Video-track (VT) locomotion, T-maze, prepulse inhibition (PPI), fear conditioning data for individual mice.
(XLSX)

**S2 Table. Raw behavioral data of the large cohort.** Video-track (VT) locomotion, T-maze, prepulse inhibition (PPI), fear conditioning data for individual mice.
(XLSX)

**S3 Table. Raw behavioral data of the replication cohort.** Video-track (VT) locomotion, T-maze, prepulse inhibition (PPI), fear conditioning data for individual mice.
(XLSX)

## Author Contributions

**Conceptualization:** Xianjin Zhou.

**Data curation:** William Yue, Sorana Caldwell, Xianjin Zhou.

**Formal analysis:** William Yue, Victoria Risbrough, Susan Powell, Xianjin Zhou.

**Funding acquisition:** Xianjin Zhou.

**Investigation:** William Yue, Sorana Caldwell, Xianjin Zhou.

**Methodology:** William Yue, Sorana Caldwell, Victoria Risbrough, Susan Powell, Xianjin Zhou.

**Project administration:** William Yue, Sorana Caldwell, Xianjin Zhou.

**Resources:** Victoria Risbrough, Susan Powell, Xianjin Zhou.

**Software:** Xianjin Zhou.

**Supervision:** Xianjin Zhou.

**Validation:** Xianjin Zhou.

**Visualization:** Xianjin Zhou.

**Writing – original draft:** Xianjin Zhou.

**Writing – review & editing:** Xianjin Zhou.

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
