## [Decision Letter · Decision Letter 0]

6 Aug 2021

PONE-D-21-23141

Chronic Presence of Blood Circulating Anti-NMDAR1 Autoantibodies Impairs Cognitive Function in Mice

PLOS ONE

Dear Dr. Zhou,

Thank you for submitting your manuscript to PLOS ONE. After careful consideration, we feel that it has merit but does not fully meet PLOS ONE’s publication criteria as it currently stands. Therefore, we invite you to submit a revised version of the manuscript that addresses the points raised during the review process.

The two experts addressed several concerns about your manuscript. Please revise your manuscript carefully.

We look forward to receiving your revised manuscript.

Kind regards,

Kenji Hashimoto, PhD

Academic Editor

PLOS ONE

Journal Requirements:

"This work was supported by the grants R21MH123705 (XZ) and

R21MH116186 (XZ) from the National Institute of Mental Health. "

"This work was supported by the grants R21MH123705 (XZ) and R21MH116186 (XZ) from the National Institute of Mental Health."

Reviewers' comments:

Reviewer's Responses to Questions

**Comments to the Author**

1. Is the manuscript technically sound, and do the data support the conclusions?

Reviewer #1: No

Reviewer #2: Partly

2. Has the statistical analysis been performed appropriately and rigorously? 

Reviewer #1: Yes

Reviewer #2: Yes

3. Have the authors made all data underlying the findings in their manuscript fully available?

Reviewer #1: Yes

Reviewer #2: Yes

4. Is the manuscript presented in an intelligible fashion and written in standard English?

Reviewer #1: Yes

Reviewer #2: Yes

5. Review Comments to the Author

Reviewer #1: This study addressed whether anti-NMDA receptor autoantibodies against a particular epitope could produce specific behavioral phenotypes. In a pilot study, a 20-amino acid-long peptide carrying a single epitope, P1 or P2, which are found in NMDAR1 subunit, was immunized with CFA in B6 male mice. Five mice out of nine resulted in the title elevation against the peptide 2. Antibody specificity was confirmed by a cell-based assay (Euroimmun) where NMDAR1 protein is expressed in HEK293 cells, and the titers’ ranges are from 1:10 to 1:100. All the mice were healthy, but those five mice displayed a deficit in fear memory extinction and T-maze spontaneous alternation is marginally defective (p=0.06), but no other consistent behavioral phenotypes were detected in locomotor activity and prepulse inhibition (PPI). The authors tried to confirm the result by immunizing P2 peptide into two independent cohort of mice. They found that the deficit in T-maze spontaneous alternation, but not fear memory extinction, is more severely impaired in the subsequent cohorts of mice. Finally, they examined the NMDAR1, GluR1 and GFAP levels of immunostaining in the cortex, hippocampus and striatum in immunized mice and control mice, and found no difference in all three proteins between the immunized mice and control mice. The authors concluded that chronic presence of the blood circulating anti-NMDAR1 autoantibodies is sufficient to specifically impair T-maze spontaneous alternation.

It is notable that mice carrying the P2 peptide-specific autoantibodies display consistent behavioral phenotypes (only T-maze alternation deficit with no other major phenotypes). This might imply that autoantibodies binding to the same epitope would share the biological effects. However, there are a few major issues in technical aspect and in manuscript writing.

Major points:

1. Fig 2 showed that antibodies generated in mice after immunization with P2 peptide bind to the naive NMDAR1 protein that are expressed in HEK203 cells by transfection of cDNA. However, it is unclear whether the antibodies are specific to NR1 protein and no other cross linked in this study. It is imperative to confirm that auto-antibody recognize only NR1 protein and no other major bands on the Western blotting. If other Mol. Weight bands did appear on the blot, I do not think auto-antibody against NR1 is “sufficient” to impair behavioral phenotypes.

2. The authors claimed in Abstract and main text that presence of anti-NMDAR1 auto-antibody is sufficient to specifically impair T-maze spontaneous alternation. However, the first cohort of mice in a pilot study, they were also clearly impaired in the fear memory extinction. I do not think T-maze is “specifically” impaired. Please remove this word.

3. No description about the consequence of using P1 peptide. Then why P1 was described in the Method? Please clarify.

4. T-maze alternation is defective although no immunohistological abnormality was detected in the staining in the striatum hippocampus and cortex in Fig 11. Medial PFC is also crucial for short-term memory in T-maze alternation. It is suggested to examine the integrity of mPFC.

Reviewer #2: In contrast to the high titers of anti-NMDAR1 autoantibodies in anti-NMDAR1 encephalitis patients, low titers of the autoantibodies are reported in general human population and psychiatric patients. The effects of this low-titer autoantibodies on human cognition and psychiatric symptoms are not clear. In this manuscript, authors established the mouse model with chronic presence of low titers of anti-NMDAR1 autoantibodies against 20 amino acids peptide derived from ligand binding domain of the NMDAR1 subunit and examined mouse cognitive and motor behaviors. Although some behavioral phenotypes observed in the 1st experiment were not detected in the 2nd and 3rd large cohorts, the authors reproducibly found the impairments in T-maze spontaneous alternation test in the mouse with the autoantibodies. Although this manuscript is very important and has some merits, the authors need to address the following points.

1) The authors used P2 peptide derived from ligand-binding domain of mouse NMDAR1 protein for immunization. Why did authors select this peptide? Does this peptide sequence locate on the surface of the NMDAR1 subunit molecule? The crystal structure of the subunit was reported, thus the rationale of the selected sequence is necessary.

2) The quantification of the titer of the anti-NMDAR1 autoantibody has some problems. The evaluation using immunohistochemical analysis is semi-quantitative. The authors used differential staining intensities between hippocampal CA1 and corpus callosum with autoantibodies as shown in Fig. 1C and 1D. In the 2nd and 3rd experiments with large cohort, the immunization protocol is different from the 1st experiment with small cohort. Furthermore, in the 3rd cohort, the authors evaluate the presence of the autoantibodies with the different in vitro One-Step quick assay. Is there any consistency of the titers of autoantibodies between immunohistochemical method and in vitro method? If the authors used same evaluation method of the titers of autoantibody in the three cohorts, the authors could analyze the relationship between the titer of the autoantibodies and behavior phenotypes in each mouse.

3) Did the autoantibody against the P2 peptide specifically recognize the NMDAR1 subunit in the mouse brain? The brain section derived from the NMDAR1-KO mouse is the ideal negative control. Or, do the autoantibodies detect only NMDAR1 subunit with western blot analysis?

Minor points

1) In the “Materials and Methods”, the 2nd to 4th lines in the “Fear conditioning” section need to be corrected to easy understand.

2) In the “Results”, statistical analysis of effect size shown in the Fig 6C is d = 1.296, but in the text “effect size (d = 1.29)” . The description of (d = 1.30) is better.

3) In the Figure Legend of Figure1 (A), CTD is not shown in Fig.1.

4) In the Figure Legend of the title of Figure 4, not only “Fear extinction”.

5) Fig. 2 and 8B, size markers are helpful.

6. PLOS authors have the option to publish the peer review history of their article (what does this mean?). If published, this will include your full peer review and any attached files.

Reviewer #1: No

Reviewer #2: No

---

## [Author Response · Author response to Decision Letter 0]

10 Aug 2021

I appreciate a thorough review, enthusiasm, and constructive suggestions from both the Editor and the reviewers. Accordingly, I have incorporated these suggestions and critiques into a revised manuscript. I am now addressing all the concerns the reviewers raised.

Reviewer 1

1. Fig 2 showed that antibodies generated in mice after immunization with P2 peptide bind to the naive NMDAR1 protein that are expressed in HEK203 cells by transfection of cDNA. However, it is unclear whether the antibodies are specific to NR1 protein and no other cross linked in this study. It is imperative to confirm that auto-antibody recognize only NR1 protein and no other major bands on the Western blotting. If other Mol. Weight bands did appear on the blot, I do not think auto-antibody against NR1 is “sufficient” to impair behavioral phenotypes.

A). The anti-NMDAR1 autoantibodies are produced by active immunization of mouse NMDAR1 peptide P2 with the CFA. The anti-NMDAR1 autoantibodies recognize NMDAR1 proteins that have a typical expression pattern in mouse hippocampus. Mice immunized only with the CFA do not produce anti-NMDAR1 autoantibodies and do not have staining in mouse hippocampus.

B). The specificity of the anti-NMDAR1 autoantibodies was further confirmed by peptide blocking experiment in Fig1D. Only the mouse NMDAR1 P2 peptide can block the binding of the autoantibodies to the NMDAR1 proteins in mouse hippocampus, whereas the NMDAR1 P1 peptide and BSA have no blocking activity at all.

C). We purchased the BIOCHIP from Euroimmun to further validate the binding of the anti-NMDAR1 autoantibodies to native NMDAR1 proteins expressed on HEK293 cells transfected with the NMDAR1 gene only. This cell-based assay is a standard diagnostic test to detect anti-NMDAR1 autoantibodies for anti-NMDAR1 encephalitis. We further conducted co-immunocytochemical staining to demonstrate a complete co-localization between known human anti-NMDAR1 autoantibody and the mouse anti-NMDAR1 autoantibodies against the P2 peptide (Fig 2B). Unlike Western blot for detection of denatured NMDAR1 proteins, the co-immunocytochemical analysis is superior in specificity.

D). In One-Step assay (Fig 8B), the anti-NMDAR1 autoantibodies were incubated with either the NMDAR1 P2 peptide tagged with GFP or NMDAR1-GFP fusion proteins that were produced and purified from E coli using 6His purification system. The anti-NMDAR1autonatibodies specifically bind both the P2-GFP and NMDAR1-GFP proteins, but not GFP proteins. 

2. The authors claimed in Abstract and main text that presence of anti-NMDAR1 auto-antibody is sufficient to specifically impair T-maze spontaneous alternation. However, the first cohort of mice in a pilot study, they were also clearly impaired in the fear memory extinction. I do not think T-maze is “specifically” impaired. Please remove this word.

We removed “specifically’ from the manuscript. However, the fear conditioning phenotypes in the small pilot mouse cohort cannot be replicated in the subsequent two large cohorts. These phenotypes are still questionable.

3. No description about the consequence of using P1 peptide. Then why P1 was described in the Method? Please clarify.

The P1 peptide was used as a control peptide for the P2 peptide blocking experiments to demonstrate the specificity of the anti-NMDAR1 autoantibodies in Fig 1D. 

4. T-maze alternation is defective although no immunohistological abnormality was detected in the staining in the striatum hippocampus and cortex in Fig 11. Medial PFC is also crucial for short-term memory in T-maze alternation. It is suggested to examine the integrity of mPFC.

We agree with the reviewer that our next goal will be to examine whether we may find any reduction of NMDAR1 proteins in mPFC.

Reviewer 2

1) The authors used P2 peptide derived from ligand-binding domain of mouse NMDAR1 protein for immunization. Why did authors select this peptide? Does this peptide sequence locate on the surface of the NMDAR1 subunit molecule? The crystal structure of the subunit was reported, thus the rationale of the selected sequence is necessary.

We provided the rationale for the selection of the peptide antigens in Materials:

“Selection of the peptides was based on immunogenicity (http://www.cbs.dtu.dk/services/NetMHCIIpan/), solubility, and surface localization on NMDAR1 proteins” 

2) The quantification of the titer of the anti-NMDAR1 autoantibody has some problems. The evaluation using immunohistochemical analysis is semi-quantitative. The authors used differential staining intensities between hippocampal CA1 and corpus callosum with autoantibodies as shown in Fig. 1C and 1D. In the 2nd and 3rd experiments with large cohort, the immunization protocol is different from the 1st experiment with small cohort. Furthermore, in the 3rd cohort, the authors evaluate the presence of the autoantibodies with the different in vitro One-Step quick assay. Is there any consistency of the titers of autoantibodies between immunohistochemical method and in vitro method? If the authors used same evaluation method of the titers of autoantibody in the three cohorts, the authors could analyze the relationship between the titer of the autoantibodies and behavior phenotypes in each mouse.

The immunization protocol is the same across all 3 mouse cohorts. The only difference is the change of Loss-of-Resistance (LOS) syringes that makes emulsification more effortless in the large mouse cohorts.

In the 3rd cohort, we used One-Step Quick assay to screen the anti-NMDAR1 autoantibodies. The One-Step quick assay is 100% consistent with the IHC analysis. Since the One-Step assay uses GFP for detection, it was used as a qualitative test. The reviewer raises an important question as to whether T-maze phenotypes may correlate to the titers of the anti-NMDAR1 autoantibodies. We did not collect mouse blood immediately after T-maze test because we did not want blood collection to complicate subsequent behavioral tests. Since the levels of blood anti-NMDAR1 autoantibodies can change significantly, the titers of the later-collected anti-NMDAR1 autoantibodies cannot be used for such correlation analysis. In the future, we need a much large mouse cohort to specifically address this question. 

3) Did the autoantibody against the P2 peptide specifically recognize the NMDAR1 subunit in the mouse brain? The brain section derived from the NMDAR1-KO mouse is the ideal negative control. Or, do the autoantibodies detect only NMDAR1 subunit with western blot analysis?

The specificity of the anti-NMDAR1 autoantibodies has been demonstrated in the peptide blocking experiments in Fig1D. The specificity of the anti-NMDAR1 autoantibodies has been addressed in the first question of the Reviewer 1.

Minor points

1) In the “Materials and Methods”, the 2nd to 4th lines in the “Fear conditioning” section need to be corrected to easy understand.

We clarified the test and added one more reference.

2) In the “Results”, statistical analysis of effect size shown in the Fig 6C is d = 1.296, but in the text “effect size (d = 1.29)” . The description of (d = 1.30) is better.

Corrected with d=1.296 in the text.

3) In the Figure Legend of Figure1 (A), CTD is not shown in Fig.1.

CTD was added.

4) In the Figure Legend of the title of Figure 4, not only “Fear extinction”.

“Fear extinction” was changed into “Fear Conditioning”.

5) Fig. 2 and 8B, size markers are helpful.

Bars were added.

---

## [Decision Letter · Decision Letter 1]

20 Aug 2021

Chronic Presence of Blood Circulating Anti-NMDAR1 Autoantibodies Impairs Cognitive Function in Mice

PONE-D-21-23141R1

Dear Dr. Zhou,

We’re pleased to inform you that your manuscript has been judged scientifically suitable for publication and will be formally accepted for publication once it meets all outstanding technical requirements.

Kind regards,

Kenji Hashimoto, PhD

Section Editor

PLOS ONE

Additional Editor Comments (optional):

Reviewers' comments:

Reviewer's Responses to Questions

**Comments to the Author**

1. If the authors have adequately addressed your comments raised in a previous round of review and you feel that this manuscript is now acceptable for publication, you may indicate that here to bypass the “Comments to the Author” section, enter your conflict of interest statement in the “Confidential to Editor” section, and submit your "Accept" recommendation.

Reviewer #1: All comments have been addressed

Reviewer #2: All comments have been addressed

2. Is the manuscript technically sound, and do the data support the conclusions?

Reviewer #1: Yes

Reviewer #2: Yes

3. Has the statistical analysis been performed appropriately and rigorously? 

Reviewer #1: Yes

Reviewer #2: Yes

4. Have the authors made all data underlying the findings in their manuscript fully available?

Reviewer #1: Yes

Reviewer #2: Yes

5. Is the manuscript presented in an intelligible fashion and written in standard English?

Reviewer #1: Yes

Reviewer #2: Yes

6. Review Comments to the Author

Reviewer #1: It is advised to ensure all the points in details regarding the specificity of autoantibodies are described in the Method section.

I have no more concerns as reviewer.

Reviewer #2: The authors have addressed and responded to all my questions and comments. The manuscript has improved well.

7. PLOS authors have the option to publish the peer review history of their article (what does this mean?). If published, this will include your full peer review and any attached files.

Reviewer #1: No

Reviewer #2: No

---

## [Editor Report · Acceptance letter]

24 Aug 2021

PONE-D-21-23141R1 

Chronic Presence of Blood Circulating Anti-NMDAR1 Autoantibodies Impairs Cognitive Function in Mice 

Dear Dr. Zhou:

I'm pleased to inform you that your manuscript has been deemed suitable for publication in PLOS ONE. Congratulations! Your manuscript is now with our production department. 

Kind regards, 

on behalf of

Prof. Kenji Hashimoto 

Section Editor

PLOS ONE